# Factors Associated with the Patient’s Decision to Avoid Healthcare during the COVID-19 Pandemic

**DOI:** 10.3390/ijerph182413239

**Published:** 2021-12-15

**Authors:** Patrícia Soares, Andreia Leite, Sara Esteves, Ana Gama, Pedro Almeida Laires, Marta Moniz, Ana Rita Pedro, Cristina Mendes Santos, Ana Rita Goes, Carla Nunes, Sónia Dias

**Affiliations:** 1Comprehensive Health Research Center, Universidade NOVA de Lisboa, Campo Mártires da Pátria 130, 1169-056 Lisboa, Portugal; andreia.leite@ensp.unl.pt (A.L.); sara.navalho@gmail.com (S.E.); ana.gama@ensp.unl.pt (A.G.); laires.pedro@gmail.com (P.A.L.); am.moniz@ensp.unl.pt (M.M.); rita.pedro@ensp.unl.pt (A.R.P.); cristina.mendes.santos@liu.se (C.M.S.); ana.goes@ensp.unl.pt (A.R.G.); CNunes@ensp.unl.pt (C.N.); sonia.dias@ensp.unl.pt (S.D.); 2NOVA National School of Public Health, Public Health Research Center, Universidade NOVA de Lisboa, Av. Padre Cruz, 1600-560 Lisboa, Portugal

**Keywords:** healthcare avoidance, health services, COVID-19, risk perception

## Abstract

The COVID-19 pandemic has resulted in changes in healthcare use. This study aimed to identify factors associated with a patient’s decision to avoid and/or delay healthcare during the COVID-19 pandemic. We used data from a community-based survey in Portugal from July 2020 to August 2021, “COVID-19 Barometer: Social Opinion”, which included data regarding health services use, risk perception and confidence in health services. We framed our analysis under Andersen’s Behavioural Model of Health Services Use and utilised Poisson regression to identify healthcare avoidance associated factors. Healthcare avoidance was high (44%). Higher prevalence of healthcare avoidance was found among women; participants who reported lower confidence in the healthcare system response to COVID-19 and non-COVID-19; lost income during the pandemic; experienced negative emotions due to physical distancing measures; answered the questionnaire before middle June 2021; and perceived having worse health, the measures implemented by the Government as inadequate, the information conveyed as unclear and confusing, a higher risk of getting COVID-19, a higher risk of complications and a higher risk of getting infected in a health institution. It is crucial to reassure the population that health services are safe. Health services should plan their recovery since delays in healthcare delivery can lead to increased or worsening morbidity, yielding economic and societal costs.

## 1. Introduction

The World Health Organization (WHO) declared the novel coronavirus disease 2019 (COVID-19) a global pandemic on 11 March 2020 [1]. Restrictive measures were implemented to contain the pandemic, such as lockdowns, stay-at-home orders, movement restriction and closure of schools and non-essential businesses [2,3]. Additionally, several countries temporarily cancelled non-urgent medical activity to ensure the best care for COVID-19 cases, diverting attention from non-COVID-19 care and a reduction in care for these conditions [2,3,4,5]. The health services reorganisation might partially explain this reduction to respond to COVID-19, but the reduction might also be explained by the patient’s avoidance or delay regarding attending healthcare due to the fear of getting COVID-19. The latter phenomenon is known as healthcare avoidance and was previously described in response to a traumatic or threatening situation like this pandemic [6,7]. Healthcare avoidance can be characterised as cancelling appointments, nonadherence to treatment and delaying or avoiding medical care due to fear or denial of symptoms and diagnosis, among other factors [6,8,9]. The impact of COVID-19 is yet to be fully determined, but existing evidence suggests that changes in healthcare utilisation are in the pathway of COVID-19 indirect effects [3].

The frequency of healthcare avoidance during the pandemic and potential drivers were analysed in previous studies [4,7,8,10,11,12]. The proportion of individuals who avoid or delay healthcare has increased in several countries, regardless of the COVID-19 incidence rate of the country, indicating a global problem. The impact of healthcare avoidance on health outcomes has not yet been fully described, although excess non-COVID-19 mortality was observed in the past few months, which could be explained by healthcare avoidance [13,14,15]. Findling et al. [10] reported that more than half of those avoiding healthcare in the USA experienced negative health consequences, underlining the importance of understanding which factors may be associated with healthcare avoidance. Reasons to avoid healthcare included the closure of medical offices, fear of contracting COVID-19 and financial difficulties resulting from the pandemic [7,10]. Other factors related to healthcare avoidance are poor health perception, the number of comorbidities and living in highly COVID-19-affected residential areas [8,11,12]. In addition, healthcare avoidance was linked to risk perception, a subjective psychological construct influenced by cognitive, emotional and cultural factors [16]. Risk perceptions are frequently subject to bias. While unrealistic optimism about health risks may result in false feelings of security and lack of precaution, pessimistic bias may lead to excessive mass scares and dissuade people from seeking healthcare [17]. Other factors, such as age, household income, education and having health insurance, were described inconsistently across the literature, suggesting that context and cultural differences in healthcare might explain these differences.

In Portugal, changes in healthcare-seeking behaviours during the pandemic were identified, with a reduction of 48% in emergency care episodes in March 2020 and 57% in all hospitalisations between March and May 2020 [18,19]. This might be due to the cancellations and difficulties accessing medical care or the patient’s decision to avoid medical care. However, healthcare avoidance has crucial consequences. Studies on previous epidemics found that individuals who avoided healthcare due to fearing infection exacerbated the severity of the disease and indirectly increased mortality through reduced access to treatment [20,21]. Furthermore, return to pre-pandemic activity levels might take a few years [21,22]. Thus, understanding the extent and factors associated with avoidance could be useful to support the development of interventions to address it [17]. Our aim was to identify factors associated with the patient’s decision to avoid and/or delay healthcare during the COVID-19 pandemic in Portugal.

## 2. Materials and Methods

### 2.1. Study Design

We used data from the online community-based survey “COVID-19 Barometer: Social Opinion”. Participants were invited using a snowball sampling approach. Invitations to participate have been sent to contact networks, mailing lists, digital social networks, patient associations and promoted in social media networks [23,24]. Participants have not received any type of compensation for participating in the questionnaire. The survey contains information on risk perception, health status, social experiences and the use of health services during the COVID-19 pandemic. Questions about sociodemographic characteristics are presented at the end of the questionnaire. Participants can answer the questionnaire once or more than once. Data collection is still ongoing, with more than 200,000 answers by August 2021. The questionnaire is flexible to adjust rapidly to the different phases and information needs throughout the pandemic. Our study started on 25 July 2020 when we added the question about healthcare avoidance and ended on 6 August 2021 when the question was removed.

We excluded participants younger than 18 years old and who were not living in Portugal. To identify questionnaires belonging to the same participant, we created a unique code based on the participant’s personal information—birthday, the mother’s birthday, residence region, educational level and sex. For participants who responded more than once, we considered the responses for the last questionnaire, except for those who answered negatively about healthcare avoidance in the last questionnaire and previously reported avoidance or delay. For these, we considered the last positive answer.

### 2.2. Variables

The dependent variable considered was the question “During the pandemic, have you avoided scheduling, or have you postponed, appointments and/or non-urgent treatments due to fear of contracting COVID-19 in health institutions?”. The answer “yes” was considered a proxy for healthcare avoidance and/or delay, referred to as healthcare avoidance hereafter.

We based our analysis on Andersen’s Behavioural Model of Health Services Use, which identified three main types of factors deemed to influence health service use: (i) predisposing, which included sociodemographic characteristics and health beliefs, namely, attitudes and values regarding health and health services; (ii) enabling, encompassing financing and organisational factors; and (iii) need for care, which incorporated individual and contextual predictors, such as the perceived need for health services and evaluated need and epidemiological indicators of morbidity [2,9,25]. Thus, we grouped the variables deemed as potential determinants of healthcare avoidance into four dimensions: predisposing, enabling, need for care and COVID-19 specific (Table 1). We also created a time variable corresponding to the questionnaire period to account for possible changes in risk perception according to the epidemiological situation of the country. Figure 1 represents the six different periods in the pandemic until August 2021, representing the different epidemic waves and inter-wave periods. In our study, five periods were considered, from P2 to P6.

All variables were collected when the answer on the dependent variable/outcome was considered. Two questions were added and/or removed during the study period. On 24 October 2020, we added a question about the perception of information provided by health authorities. On 14 May 2021, this question and the question regarding the perceived risk of getting infected in a health institution were removed. Individuals who did not answer these questions were included in the analysis but had a missing value for these variables. We performed a complete case analysis with pairwise deletion. Except for the two questions that were only available for a few weeks during the study, all the variables had less than 2.5% missing data. Monthly household income was the only variable with 10% missing data.

### 2.3. Statistical Analysis

Variables were described using absolute and relative frequencies. Logistic regression is usually used in cross-sectional studies with binary outcomes, which estimates odds ratios (ORs). However, ORs are overestimated in the presence of frequent events. Thus, given the high frequency of healthcare avoidance, we fitted Poisson regression models with robust standard errors using sandwich estimation [26]. Prevalence ratios (PRs) and the corresponding 95% confidence intervals (CIs) were estimated for each variable. PRs were adjusted for the predisposing factors (gender, age group, region and education) and the need for care factors for health services use (health perception and period of the questionnaire) [9,25].

All statistical analyses were performed using R 4.0.2 [27].

## 3. Results

Between July 2020 and August 2021, 9660 individuals participated and were included in the analysis. Table 2 presents the characteristics of the sample. Overall, more women (74.1%), individuals who lived in Lisbon and Tagus Valley (55.4%), working-age adults (80.2%) and individuals with higher education (71.5%) answered the questionnaire. In total, 43.6% (*n* = 4216) of the participants stated having avoided or delayed healthcare during the pandemic.

Crude PRs and adjusted PRs (aPRs) with their respective 95% CIs can be found in the Appendix A (Appendix A).

### 3.1. Predisposing Factors

The prevalence of avoiding healthcare was higher for women than men (aPR: 1.27, 95% CI: 1.20–1.35) (Figure 2). A higher prevalence of healthcare avoidance was also found for individuals with low confidence in the responses of health services to COVID-19 and non-COVID-19 compared with individuals reporting higher trust in the health services (aPR: 1.19, 95% CI: 1.13–1.25 and aPR: 1.24, 95% CI: 1.18–1.30, respectively) (Figure 2). Young adults and individuals who reported living in the North, Center or Azores had a lower prevalence of healthcare avoidance than working-age adults and individuals who reported living in Lisbon and Tagus Valley (Figure 2 and Appendix A).

### 3.2. Enabling Factors

The prevalence of healthcare avoidance was higher for participants who lost income during the pandemic than individuals who did not (aPR: 1.10, 95% CI: 1.04–1.15) (Figure 2).

### 3.3. Need for Care

Participants who perceived their health status as reasonable or bad had a higher prevalence of healthcare avoidance than participants who perceived their health as good (aPR: 1.25, 95% CI: 1.19–1.31 and aPR: 1.38, 95% CI: 1.23–1.54, respectively) (Figure 3). Similarly, participants with one disease had a higher prevalence of healthcare avoidance than participants without diseases (aPR: 1.06, 95% CI: 1.01–1.12) (Figure 3). Participants who frequently reported feeling agitated, sad or anxious due to physical distancing measures were also more likely to avoid healthcare than participants who never experienced those feelings (Figure 3 and Appendix A). The prevalence of healthcare avoidance was also higher during the initial periods of the pandemic than in the last months (Figure 3 and Appendix A).

### 3.4. COVID-19-Specific Factors

Participants who perceived their risk of getting COVID-19 as low or non-existent had a lower prevalence of healthcare avoidance than those who perceived a high risk (aPR: 0.88, 95% CI: 0.82–0.95) (Figure 3). Similarly, participants who perceived their risk of having complications following a diagnosis of COVID-19 as low or non-existent or were unsure had a lower prevalence of healthcare avoidance than those who perceived their risk as high (aPR: 0.82, 95% CI: 0.77–0.88 and aPR: 0.91, 95% CI: 0.84–0.99, respectively) (Figure 3). Participants who perceived a high risk of infection in a health institution also had a higher prevalence of avoiding healthcare than participants who perceived their risk as moderate, low or unsure (Figure 3 and Appendix A). Participants who had an inadequate perception of the level of adequacy of the measures implemented by the Government had a higher prevalence of healthcare avoidance than those who had an adequate perception (aPR: 1.07, 95% CI: 1.02–1.12) (Figure 3). Additionally, participants who found the information provided by the health authorities as unclear and confusing had a higher prevalence of healthcare avoidance than those who found the information clear and understandable (aPR: 1.11, 95% CI: 1.03–1.21) (Figure 3 and Appendix A).

## 4. Discussion

This study found that almost 44% of the participants avoided or delayed healthcare during the pandemic due to fear of contracting COVID-19 in health institutions in Portugal. We found a higher prevalence of healthcare avoidance in women and participants with low confidence in the health response to COVID-19 and non-COVID-19 conditions. Likewise, a higher prevalence of healthcare avoidance was reported by those who lost income during the pandemic; did not perceive their health as good; and experienced sadness, anxiety or agitation due to physical distancing measures. Those who answered the questionnaire before middle June 2021 (P6), found the measures implemented by the Government inadequate, and perceived the information reported as unclear and confusing also reported a higher prevalence of healthcare avoidance. In contrast, a lower prevalence of healthcare avoidance was found in participants who perceived their risk of getting infected, getting infected at a health institution and having complications as low or non-existent.

The prevalence of healthcare avoidance is widely variable within and between countries, with studies in the United States, Australia and South Korea reporting estimates between 15 and 73%, and 10 to 12% avoiding urgent care and around 32% avoiding routine care [8,10,11,12,28]. Studies exploring the factors associated with healthcare avoidance during the COVID-19 pandemic are still scarce. However, some factors seem consistent across the literature. We found that the prevalence of healthcare avoidance was higher for participants who did not perceive their health as good; had one disease; and experienced frequent feelings of anxiety, sadness and agitation. These results are in agreement with the literature, with a higher prevalence of healthcare avoidance for individuals with poor health perception, comorbidities, disabilities and symptoms of anxiety and depression [11,12,28,29]. Although participants with two or more diseases did not have higher healthcare avoidance than participants without diseases in our study, this is likely due to the adjustment with the perception of health status. A higher prevalence of healthcare avoidance in women was also found in the literature [8,11,28]. Behavioural differences might explain this finding since previous studies found that women perceived themselves as more vulnerable to illnesses, were more likely to use health services during pain episodes and, in general, visited general practitioners more often than men [30,31]. Higher odds of healthcare avoidance were found for individuals who lived in highly COVID-19-affected areas [8]. Although we did not distinguish between the regional incidence in each phase of the pandemic, we explored healthcare avoidance according to different pandemic phases and found higher healthcare avoidance at the beginning and during a high-incidence period, indicating the importance of the overall situation perception.

Participants who perceived their risk of getting COVID-19 and complications as low or non-existent had a lower prevalence of healthcare avoidance than those who perceived their risk as high. However, this finding was not replicated in one study that dichotomised the self-perceived risk of having severe COVID-19 [12], suggesting that the different results obtained among different studies might be due to methodological differences. Our questions regarding risk perception were not binary but ordinal (high, moderate, low/non-existent and unsure), similar to another study that used a five-level scale to assess risk perceptions in different groups: to the community, infection for someone in their age group and complications in case of infection. The latter found results similar to ours [28]. Individuals with higher risk perceptions also had lower trust in their governments [16]. Similarly, we found a higher prevalence of healthcare avoidance for individuals who had low trust in health services and perceived the measures implemented by the Government during the pandemic as inadequate. Although these results suggest that trust might influence risk perception, the literature is unclear regarding this relationship and its correlation with attitudes, especially during outbreaks [32].

The results are inconclusive regarding age, income and education [8,11,12,28]. Although we did not find differences in the prevalence of healthcare avoidance for household income and education, participants who lost income during the pandemic had a higher prevalence than those who did not.

Our study had some limitations. It is not representative of the Portuguese population since we had an overrepresentation of women, individuals with a university degree and from Lisbon and Tagus Valley. Thus, it is possible that the prevalence of healthcare avoidance might be overestimated since the prevalence was higher for women and Lisbon and Tagus Valley than other regions of Portugal. Additionally, the homeless and users with limited internet access were likely underrepresented and users without a social network profile or email account and/or with limited digital literacy or IT skills might not have even come across the questionnaire. Our study also suffered from response and nonresponse bias since participants might have given more socially acceptable answers, though anonymised online surveys minimise this effect. Individuals might also answer that they avoided healthcare but did it for other reasons than fearing COVID-19. Nevertheless, we found higher healthcare avoidance prevalence and a high risk perception of getting COVID-19 in a health institution, which could be a sign of consistency across the questionnaire. Volunteer bias may have also been present, as participants who feared and were more concerned with COVID-19 or its severity might have been more likely to answer the questionnaire, which might have underestimated the proportion of individuals who perceived their risk of getting COVID-19 and complications as low or non-existent, and possibly overestimated the prevalence of healthcare avoidance. Although we created a unique code for each participant based on the participant’s personal information, there was still a possibility of repeated sampling since participants could share a birthday, live in the same region and have the same education level. Finally, the number of diseases may have been underestimated as the prevalence of chronic diseases reported was lower than that reported in the National Health Survey [33]. We also do not have detailed information to distinguish between individuals who needed urgent care from those who did not, nor the reason to need healthcare. Fear of contracting COVID-19 in health institutions could have different meanings for each participant. Although an interesting question, this was not addressed in our study. The perception of needing healthcare is also subjective, and one might delay or avoid checking on symptoms that may lead to serious conditions. In contrast, someone else might avoid or delay appointments deemed less important or for preventive reasons. It is also possible that a participant answered “yes” and avoided healthcare but went to the emergency care instead of scheduling an appointment at the primary care level.

Nevertheless, it is still alarming that participants with perceived poor health status had a higher prevalence of healthcare avoidance. Further research would be valuable to explore the reasons to avoid healthcare, e.g., through a qualitative approach. Additionally, it would be interesting to further explore the reasons behind a low trust in the health services’ responses to COVID-19 and non-COVID-19 to understand how to improve confidence in health services.

Our study also had various strengths. We analysed a large number of participants who answered questions concerning several distinct dimensions connected with health services use, which allowed us to consider a broad overview of the several dimensions associated with healthcare use. Additionally, our questionnaire has been online for more than one year now, which is helpful to compare different time points and how behaviours have changed over time. Thus, our study provides a broad snapshot of healthcare avoidance in Portugal during the COVID-19 pandemic.

Our results point to two main areas of action: quantification of one’s risk of SARS-CoV-2 infection and complications and recovery of the lost activity. Governments and health authorities should help the population accurately assess and quantify their risk of infection and severe COVID-19 and ensure that health services are safe. Fear might lead to higher anxiety and stress levels, overestimating the risk and biased risk perceptions [17,34]. Additionally, it is essential to decrease healthcare avoidance since avoiding medical care might lead to delays in diagnosis, increased morbidity and mortality [20]. Therefore, health services must plan to recover the lost activity since several appointments, treatments, surgeries and medical exams were cancelled, delayed and avoided, and waiting lists have increased. Telemedicine could be a part of the solution. Nevertheless, health services will likely need to increase their activity levels to recover the lost activity, translating into more human resources, which must be accounted for in future governmental budgets.

## 5. Conclusions

We found a high prevalence of healthcare avoidance due to the fear of contracting COVID-19 in health institutions. Overall, our results indicated that authorities should ensure that health services are safe and encourage utilisation. Additionally, authorities should help the population accurately assess and quantify their risk of infection to decrease healthcare avoidance based on fear. We have a long road ahead of us to recover the activity lost during the pandemic, either due to the medical care being cancelled or the patient’s decision to delay and/or avoid health services due to fearing a SARS-CoV-2 infection. Health services should carefully plan their recovery, as healthcare avoidance might lead to increased morbidity and mortality.

## Figures and Tables

**Figure 1 ijerph-18-13239-f001:**
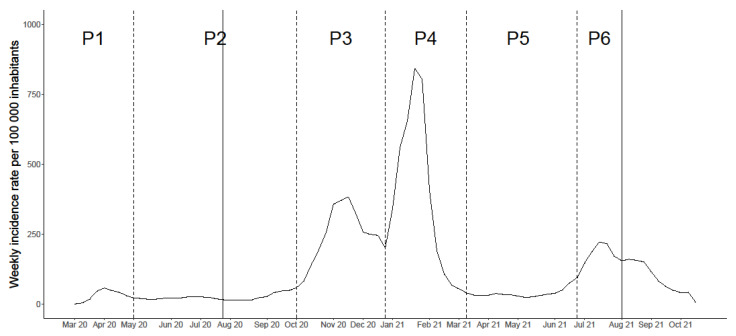
Weekly incidence rate of COVID-19 in Portugal. The solid vertical lines correspond to the beginning and end of the study period, while the dotted lines represent the defined epidemic waves and inter-wave periods. P—period of the pandemic.

**Figure 2 ijerph-18-13239-f002:**
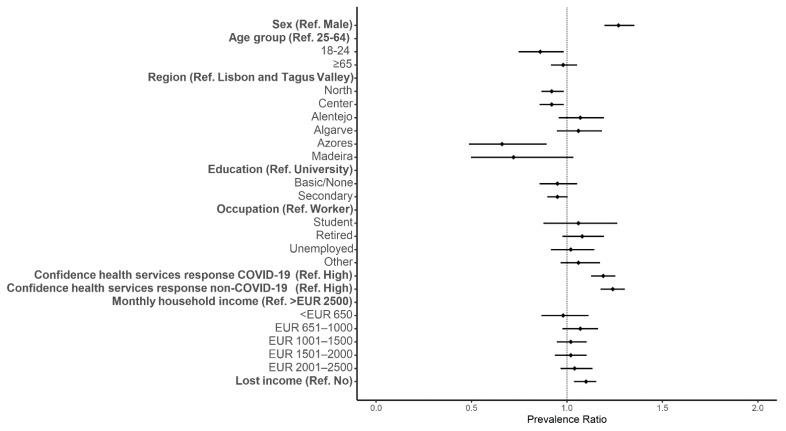
Forest plot of healthcare avoidance for predisposing and enabling factors. Adjusted prevalence ratio (adjusted for gender, age group, region, education, health perception and period of the questionnaire) and the respective 95% confidence intervals are denoted by black dots and black lines, respectively.

**Figure 3 ijerph-18-13239-f003:**
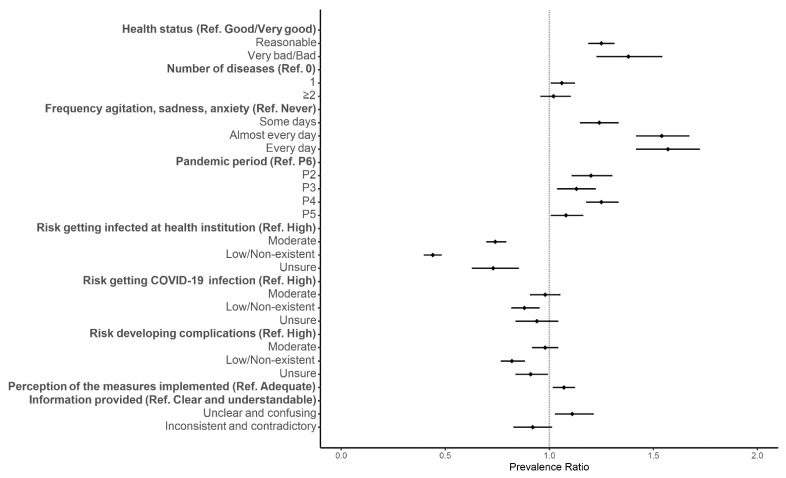
Forest plot of healthcare avoidance for the need for care and COVID-19-specific factors. Adjusted prevalence ratio (adjusted for gender, age group, region, education, health perception and period of the questionnaire) and the respective 95% confidence intervals are denoted by black dots and black lines, respectively.

**Table 1 ijerph-18-13239-t001:** Dimensions and variables considered for analysis, based on Andersen’s Behavioural Model [9,25].

Dimensions Considered	Variables
Predisposing	Sex
	Age group
	Region
	Education
	Occupation
	Confidence in the capacity of health services to respond to COVID-19
	Confidence in the capacity of health services to respond to non-COVID-19
Enabling	Monthly household income
	Partial or total income loss during the pandemic
Need for care	Perception of the health status
	Number of diseases
	Frequency of agitation, sadness or anxiety due to the physical distancing measures
	Period of the pandemic
COVID-19 specific	Self-perceived risk of getting COVID-19
	Self-perceived risk of developing severe disease following SARS-CoV-2 infection
	Self-perceived risk of getting infected in a health institution
	Perception of the level of adequacy of measures implemented by the Government
	Perception of the information provided by health authorities

**Table 2 ijerph-18-13239-t002:** Sample characteristics according to predisposing, enabling, need for care and COVID-19-specific factors.

	Total Sample (*N* = 9660)	Healthcare Avoidance (*N* = 4216, 43.6%)	Did Not Avoid and/or Delay Healthcare (*N* = 5444, 56.4%)
	*N* (%)	*N* (%)	*N* (%)
Predisposing			
Sex (*N* = 9626)			
Male	2494 (25.9%)	911 (21.7%)	1583 (29.2%)
Female	7132 (74.1%)	3289 (78.3%)	3843 (70.8%)
Age (*N* = 9660)			
18–24 years	384 (4.0%)	146 (3.5%)	238 (4.4%)
25–64 years	7752 (80.2%)	3416 (81.0%)	4336 (79.6%)
≥65 years	1524 (15.8%)	654 (15.5%)	870 (16.0%)
Region (*N* = 9660)			
North	1942 (20.1%)	806 (19.1%)	1136 (20.9%)
Center	1406 (14.6%)	583 (13.8%)	823 (15.1%)
Lisbon and Tagus Valley	5354 (55.4%)	2399 (56.9%)	2955 (54.3%)
Alentejo	417 (4.32%)	198 (4.70%)	219 (4.02%)
Algarve	381 (3.94%)	181 (4.29%)	200 (3.67%)
Azores	100 (1.04%)	30 (0.71%)	70 (1.29%)
Madeira	60 (0.62%)	19 (0.45%)	41 (0.75%)
Education (*N* = 9615)			
No education/Basic education	573 (5.96%)	241 (5.75%)	332 (6.12%)
Secondary	2166 (22.5%)	914 (21.8%)	1252 (23.1%)
University	6876 (71.5%)	3038 (72.5%)	3838 (70.8%)
Occupation (*N* = 9660)			
Worker	6849 (70.9%)	2971 (70.5%)	3878 (71.2%)
Student	339 (3.51%)	137 (3.25%)	202 (3.71%)
Retired	1434 (14.8%)	635 (15.1%)	799 (14.7%)
Unemployed	451 (4.67%)	198 (4.70%)	253 (4.65%)
Other	587 (6.08%)	275 (6.52%)	312 (5.73%)
Confidence in the capacity of health services to respond to COVID-19 (*N* = 9585)			
High	7361 (76.8%)	3068 (73.3%)	4293 (79.5%)
Low	2224 (23.2%)	1118 (26.7%)	1106 (20.5%)
Confidence in the capacity of health services to respond to non-COVID-19 (*N* = 9593)			
High	4423 (46.1%)	1688 (40.4%)	2735 (50.5%)
Low	5170 (53.9%)	2490 (59.6%)	2680 (49.5%)
Enabling			
Monthly household income (*N* = 8644)			
<EUR 650	508 (5.88%)	211 (5.61%)	297 (6.08%)
EUR 651–1000	1222 (14.1%)	553 (14.7%)	669 (13.7%)
EUR 1001–1500	1878 (21.7%)	830 (22.1%)	1048 (21.4%)
EUR 1501–2000	1587 (18.4%)	680 (18.1%)	907 (18.6%)
EUR 2001–2500	1352 (15.6%)	607 (16.2%)	745 (15.2%)
>EUR 2501	2097 (24.3%)	877 (23.3%)	1220 (25.0%)
Loss of income due to the pandemic (*N* = 9446)			
No	6778 (71.8%)	2870 (69.9%)	3908 (73.2%)
Partial/Total	2668 (28.2%)	1237 (30.1%)	1431 (26.8%)
Need for care			
Perception of the health status (*N* = 9625)			
Very good/Good	5418 (56.3%)	2121 (50.4%)	3297 (60.8%)
Reasonable	3889 (40.4%)	1914 (45.5%)	1975 (36.4%)
Bad/Very bad	318 (3.30%)	170 (4.04%)	148 (2.73%)
Number of diseases (*N* = 9413)			
0	5018 (53.3%)	2084 (50.6%)	2934 (55.5%)
1	2853 (30.3%)	1326 (32.2%)	1527 (28.9%)
≥2	1537 (16.3%)	709 (17.2%)	828 (15.7%)
Frequency of agitation, sadness or anxiety due to the physical distance measures (*N* = 9624)			
Never	1901 (19.8%)	612 (14.6%)	1289 (23.8%)
Some days	5588 (58.1%)	2402 (57.1%)	3186 (58.8%)
Almost every day	1411 (14.7%)	777 (18.5%)	634 (11.7%)
Every day	724 (7.52%)	412 (9.80%)	312 (5.76%)
Pandemic period (*N* = 9660)			
P2	1071 (11.1%)	499 (11.8%)	572 (10.5%)
P3	1121 (11.6%)	486 (11.5%)	635 (11.7%)
P4	2284 (23.6%)	1116 (26.5%)	1168 (21.5%)
P5	1757 (18.2%)	757 (18.0%)	1000 (18.4%)
P6	3427 (35.5%)	1358 (32.2%)	2069 (38.0%)
COVID-19 specific			
Self-perceived risk of getting COVID-19 (*N* = 9635)			
High	1091 (11.3%)	533 (12.7%)	558 (10.3%)
Moderate	4004 (41.6%)	1836 (43.6%)	2168 (39.9%)
Low/No risk	3885 (40.3%)	1546 (36.7%)	2339 (43.1%)
Unsure	655 (6.80%)	292 (6.94%)	363 (6.69%)
Self-perceived risk to develop severe disease following SARS-CoV-2 infection (*N* = 9627)			
High	1699 (17.6%)	864 (20.5%)	835 (15.4%)
Moderate	2948 (30.6%)	1384 (32.9%)	1564 (28.8%)
Low/No risk	3639 (37.8%)	1369 (32.6%)	2270 (41.9%)
Unsure	1341 (13.9%)	588 (14.0%)	753 (13.9%)
Self-perceived risk to get infected in a health institution (*N* = 5399)			
High	822 (15.2%)	566 (22.6%)	256 (8.84%)
Moderate	2429 (45.0%)	1247 (49.8%)	1182 (40.8%)
Low/No risk	1978 (36.6%)	604 (24.1%)	1374 (47.4%)
Unsure	170 (3.15%)	86 (3.44%)	84 (2.90%)
Perception of the level of adequacy of the measures implemented by the Government (*N* = 9423)			
Adequate	5886 (62.5%)	2509 (61.1%)	3377 (63.5%)
Inadequate	3537 (37.5%)	1597 (38.9%)	1940 (36.5%)
View on the information provided by the health authorities (*N* = 3926)			
Clear and understandable	2398 (61.1%)	1118 (61.1%)	1280 (61.1%)
Unclear and confusing	730 (18.6%)	380 (20.8%)	350 (16.7%)
Inconsistent and contradictory	798 (20.3%)	333 (18.2%)	465 (22.2%)

## Data Availability

The data presented in this study are available on request from the corresponding author. The data are not publicly available since this is an ongoing study.

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
