# Peer review of "Factors Associated with the Patient’s Decision to Avoid Healthcare during the COVID-19 Pandemic"

_ijerph, 2021, doi:10.3390/ijerph182413239_

Round 1

Reviewer 1 Report

This is an interesting paper. Yet, there are various possible channels for why you observe what you do in this study. Ideally, those should all be addressed. See my comments below.

  1. Given the question you asked, i.e., “During the pandemic, have you avoided scheduling, or have you postponed, appointments and/or non-urgent treatments due to fear of contracting COVID-19 in health institutions?”, it is unclear whether a person who answered “yes” is indeed afraid of a COVID-19 infection or whether she is only afraid of being tested positive. A positive test result does not imply that you have serious health conditions, also because there are positive test cases that do not develop any symptoms. However, being tested positive has huge consequences for individuals, such as stay-at-home orders, etc.. It would be good, if there were a way to distinguish between those two motives.
  2. There may be a Hawthorne effect in your survey. People may respond “yes” to this question simply because they don’t want to admit other reasons for health care avoidance than fear of Covid. Especially, due to a well-known overconfidence bias (e. g. Pallier et al., 2002), people may deny to themselves and to others that they are fearful.
  3. Specifically, people may avoid health care because their trust in healthcare has decreased. The Lisbon Court of Appeal ruled on November 11, 2020, that the PCR test “is unable to determine, beyond reasonable doubt, that a positive result corresponds, in fact, to the infection of a person by the SARS-CoV-2 virus.” The court based its decision, inter alia, on the study by Surkoya et al. (2020). Other studies confirming the unreliability of PCR tests are Jia et al. (2021) and Jaafar et al. (2021). Since this court decision is public but PCR tests continued to be used routinely in health care institutions, this may have led to mistrust in the quality of healthcare provided by health institutions and therefore to decreased utilization thereof.
  4. The government encouraged people to avoid healthcare while inciting people’s fear about COVID. In particular, the Portuguese government has announced that people should avoid healthcare with primary physicians and instead go to the hospital immediately in case people have symptoms. An individual who just followed those orders may respond “yes” to the question since they avoided scheduling with their physician but eventually got healthcare by going to the hospital (which was not part of the question). So we have a potential confound here.
  5. People could also have decreased use of healthcare because of altruism, i.e. they may have pity on those infected with COVID-19 and may want to give those people priority. A measure for altruism in your survey, e.g. a dictator game, may have been good.
  6. Did you consider using the probit model for the analysis?
  7. In which order did you present the questionnaire? Did participants first have to provide personal characteristics or were they first asked the question whether they have avoided scheduling appointments?
  8. Was participation in the survey incentivized?
  9. Do you find that response behavior is stable over time? As time progressed between 2020 and 2021, more information on COVID became available, which may have changed response behavior.

Minor comments:

  1. “Additionally, several countries … diverting attention from non-COVID-19 care. As a result, healthcare utilization for non-COVID-19 conditions decreased …” (p. 1) Why “as a result”? The decrease in healthcare utilization could have many reasons.
  2. 6: “Volunteer bias may also be present, as participants who do not believe in COVID-19 or its severity might be less likely to answer the questionnaire.” I am not sure whether you can make this statement, because these people feel particularly keen on giving feedback on the situation, which the questionnaire enables them to do.
  3. The quality in terms of pixels of figures 2 and 3 is not so good. Can you improve that?

References:

Jaafar, R., Aherfi, S., Wurtz, N., Grimaldier, C., Van Hoang, T., Colson, P., ... & La Scola, B. (2021). Correlation between 3790 quantitative polymerase chain reaction–positives samples and positive cell cultures, including 1941 severe acute respiratory syndrome coronavirus 2 isolates. Clinical Infectious Diseases72(11), e921-e921.

Jia, X., Xiao, L., & Liu, Y. (2021). False negative RT-PCR and false positive antibody tests–Concern and solutions in the diagnosis of COVID-19. Journal of Infection82(3), 414-451.

Pallier, Gerry; Wilkinson, Rebecca; Danthiir, Vanessa; Kleitman, Sabina; Knezevic, Goran; Stankov, Lazar; Roberts, Richard D. (2002). "The Role of Individual Differences in the Accuracy of Confidence Judgments". The Journal of General Psychology129 (3): 257–299.

Surkova, E., Nikolayevskyy, V., & Drobniewski, F. (2020). False-positive COVID-19 results: hidden problems and costs. The Lancet Respiratory Medicine, 8(12), 1167–1168. doi.org/10.1016/S2213-2600(20)30453-7

Author Response

This is an interesting paper. Yet, there are various possible channels for why you observe what you do in this study. Ideally, those should all be addressed. See my comments below.

Thank you for your positive and constructive comments.

Given the question you asked, i.e., “During the pandemic, have you avoided scheduling, or have you postponed, appointments and/or non-urgent treatments due to fear of contracting COVID-19 in health institutions?”, it is unclear whether a person who answered “yes” is indeed afraid of a COVID-19 infection or whether she is only afraid of being tested positive. A positive test result does not imply that you have serious health conditions, also because there are positive test cases that do not develop any symptoms. However, being tested positive has huge consequences for individuals, such as stay-at-home orders, etc.. It would be good, if there were a way to distinguish between those two motives.

Thank you for pointing out this aspect. Currently, we cannot ascertain the motives behind the fear of contracting COVID-19 in health institutions. However, we agree with the reviewer, it would be interesting to distinguish between potential reasons, possibly through qualitative research. Therefore, taking into account the reviewer’s comment, we added this aspect to the limitations and highlighted that further research would be valuable to explore the reasons to avoid healthcare. We’ve updated the discussion:

“We also do not have detailed information to distinguish between individuals who needed urgent care from those who did not, nor the reason to need healthcare. Fear of contracting COVID-19 in health institutions could have different meanings for each participant. Although an interesting question, this was not addressed in our study. (…) Further research would be valuable to explore the reasons to avoid healthcare, e.g. through a qualitative approach.”

There may be a Hawthorne effect in your survey. People may respond “yes” to this question simply because they don’t want to admit other reasons for health care avoidance than fear of Covid. Especially, due to a well-known overconfidence bias (e. g. Pallier et al., 2002), people may deny to themselves and to others that they are fearful.

Thank you for your comment. We agree that individuals answering that they avoided health care due to fear of COVID-19 infection might actually have other reasons for that. We’ve revised the discussion to address your comment and now mention that the actual cause for avoiding healthcare could be other than COVID.

“Our study also suffers from response and nonresponse bias since participants might have given more socially acceptable answers though anonymised online surveys minimise this effect. Individuals might also answer that they avoided healthcare but did it for other reasons than fearing COVID-19. Nevertheless, we found higher healthcare avoidance prevalence and high risk perception of getting COVID-19 in a health institution, which could be a sign of consistency across the questionnaire.”

Specifically, people may avoid health care because their trust in healthcare has decreased. The Lisbon Court of Appeal ruled on November 11, 2020, that the PCR test “is unable to determine, beyond reasonable doubt, that a positive result corresponds, in fact, to the infection of a person by the SARS-CoV-2 virus.” The court based its decision, inter alia, on the study by Surkoya et al. (2020). Other studies confirming the unreliability of PCR tests are Jia et al. (2021) and Jaafar et al. (2021). Since this court decision is public but PCR tests continued to be used routinely in health care institutions, this may have led to mistrust in the quality of healthcare provided by health institutions and therefore to decreased utilization thereof.

We thank the Reviewer for this comment. We found higher healthcare avoidance prevalence for individuals who did not perceive the measures implemented by the Government as adequate and had low confidence in the health services response to COVID and non-COVID. However, several reasons could be behind low confidence in the quality of healthcare. In this study, we’re unable to distinguish between the reasons behind low confidence in health services response since it was not the purpose of this study. We’ve added that this could be further explored in qualitative research:

“Further research would be valuable to explore the reasons to avoid healthcare, e.g. through a qualitative approach. Additionally, it would be interesting to further explore the reasons behind a low trust in the health services response to COVID-19 and non-COVID-19 to understand how to improve confidence in health services.”

The government encouraged people to avoid healthcare while inciting people’s fear about COVID. In particular, the Portuguese government has announced that people should avoid healthcare with primary physicians and instead go to the hospital immediately in case people have symptoms. An individual who just followed those orders may respond “yes” to the question since they avoided scheduling with their physician but eventually got healthcare by going to the hospital (which was not part of the question). So we have a potential confound here.

We agree with your comment. The interpretation of the question might lead to different responses creating bias, however, our interest was whether healthcare was avoided or not. We’ve revised the discussion:

“The perception of needing healthcare is also subjective, and one might delay or avoid checking on symptoms that may lead to serious conditions. In contrast, someone else might avoid or delay appointments deemed less important or for preventive reasons. It is also possible that a participant answered “Yes” and avoided healthcare but went to the emergency care instead of scheduling an appointment at the primary care level.” 

People could also have decreased use of healthcare because of altruism, i.e. they may have pity on those infected with COVID-19 and may want to give those people priority. A measure for altruism in your survey, e.g. a dictator game, may have been good.

Thank you for your comment. Indeed, it could have been interesting to get a measure of altruism to see how this could influence healthcare avoidance, however, reasons behind avoidance were outside the scope of our study. We’ve added a sentence on the limitations and on further research:

“The perception of needing healthcare is also subjective, and one might delay or avoid checking on symptoms that may lead to serious conditions. In contrast, someone else might avoid or delay appointments deemed less important or for preventive reasons. It is also possible that a participant answered “Yes” and avoided healthcare but went to the emergency care instead of scheduling an appointment at the primary care level. (…)Further research would be valuable to explore the reasons to avoid healthcare, e.g. through a qualitative approach.”  

Did you consider using the probit model for the analysis?

Thank you for your comment. We did not consider the probit regression for the analysis as odds-ratio (OR) are the most commonly used measure of effect in binary outcomes. However, since we have a frequent event, we fitted Poisson regression with robust standard errors and estimated prevalence ratios, which are easier to interpret than the output of the probit regression.

Zou, G. A Modified Poisson Regression Approach to Prospective Studies with Binary Data. Am. J. Epidemiol. 2004, 159 (7), 702–706. https://doi.org/10.1093/aje/kwh090.

In which order did you present the questionnaire? Did participants first have to provide personal characteristics or were they first asked the question whether they have avoided scheduling appointments?

Questions about sociodemographic characteristics were asked variables were presented at the end of the questionnaire.

Was participation in the survey incentivized?

Participants did not receive any type of compensation for participating in the questionnaire. We’ve added a sentence in the methodology:

“Participants did not receive any type of compensation for participating in the questionnaire.”

Do you find that response behavior is stable over time? As time progressed between 2020 and 2021, more information on COVID became available, which may have changed response behavior.

Thank you for your comment. This was definitely considered as we had the same rationale. Participants who answered the questionnaire at earlier times had a higher healthcare avoidance prevalence than individuals who answered the questionnaire before middle June 2021.

Minor comments:

“Additionally, several countries … diverting attention from non-COVID-19 care. As a result, healthcare utilization for non-COVID-19 conditions decreased …” (p. 1) Why “as a result”? The decrease in healthcare utilization could have many reasons.

Thank you for your comment. We agree, the decrease in healthcare utilisation has several causes, therefore we rephrased the sentence:

“Additionally, several countries temporarily cancelled non-urgent medical activity to ensure the best care for COVID-19 cases, diverting attention from non-COVID-19 care and a reduction in care for these conditions 2–5. The health services reorganisation might partially explain this reduction to respond to COVID-19, but the reduction might also be explained by the patient’s avoidance or delay attending healthcare due to fear of getting COVID-19.”

“Volunteer bias may also be present, as participants who do not believe in COVID-19 or its severity might be less likely to answer the questionnaire.” I am not sure whether you can make this statement, because these people feel particularly keen on giving feedback on the situation, which the questionnaire enables them to do.

Thank you for your comment. We reformulated the sentence:

“Volunteer bias may also be present, as participants who fear and are more concerned with COVID-19 or its severity might be more likely to answer the questionnaire, which might underestimate the proportion of individuals who perceive their risk of getting COVID-19 and complications as low or non-existent, and possibly overestimate the prevalence of healthcare avoidance.”

The quality in terms of pixels of figures 2 and 3 is not so good. Can you improve that?

Thank you for noticing. We provided figures with higher resolution.

References:

Jaafar, R., Aherfi, S., Wurtz, N., Grimaldier, C., Van Hoang, T., Colson, P., ... & La Scola, B. (2021). Correlation between 3790 quantitative polymerase chain reaction–positives samples and positive cell cultures, including 1941 severe acute respiratory syndrome coronavirus 2 isolates. Clinical Infectious Diseases72(11), e921-e921.

Jia, X., Xiao, L., & Liu, Y. (2021). False negative RT-PCR and false positive antibody tests–Concern and solutions in the diagnosis of COVID-19. Journal of Infection82(3), 414-451.

Pallier, Gerry; Wilkinson, Rebecca; Danthiir, Vanessa; Kleitman, Sabina; Knezevic, Goran; Stankov, Lazar; Roberts, Richard D. (2002). "The Role of Individual Differences in the Accuracy of Confidence Judgments". The Journal of General Psychology129 (3): 257–299.

Surkova, E., Nikolayevskyy, V., & Drobniewski, F. (2020). False-positive COVID-19 results: hidden problems and costs. The Lancet Respiratory Medicine, 8(12), 1167–1168. doi.org/10.1016/S2213-2600(20)30453-7

Reviewer 2 Report

The impact of COVID-19 on human life is of worldwide interest. In particular, the impact on access to healthcare is important because it may have a significant impact on the subsequent prognosis of patients.
Therefore, I believe that this study is very significant.

【Minor Comments】
※Modification is not mandatory

・L127-129:

 ”Individuals who did not answer those questions were included in the analysis but had a missing value for these variables.”

This may indicate the presence of missing values.
If so, I thought it would be better to add a note on how the missing values were handled.

・L254-276:

Limitations are described.

This explains the limitations.
Of particular interest to me was the sampling for data collection.
The survey was conducted as an online survey. As such, it was assumed that the respondents had a certain level of IT literacy. This implies that the respondents have a certain level of information gathering ability. Therefore, the impact of people with low information gathering ability may not be taken into account.
This point has also been shown for access to the Internet. However, I think what is important is not the access to the Internet itself, but the bias due to the accessibility of information and the bias due to the reliability of the information source.

Author Response

The impact of COVID-19 on human life is of worldwide interest. In particular, the impact on access to healthcare is important because it may have a significant impact on the subsequent prognosis of patients.
Therefore, I believe that this study is very significant.

We thank the reviewer for their comments and for reviewing our manuscript.

【Minor Comments】
※Modification is not mandatory

・L127-129:

 ”Individuals who did not answer those questions were included in the analysis but had a missing value for these variables.”

This may indicate the presence of missing values.
If so, I thought it would be better to add a note on how the missing values were handled.

Thank you for your comment. Indeed, these situations correspond to missing values, which were excluded from the analysis. We added a note in the methodology:

“We performed a complete case analysis with pairwise deletion. Except for the two questions that were only available a few weeks during the study, all the variables had less than 2.5% missing data. Monthly household income was the only variable with 10% of missing data.”

 ・L254-276:

Limitations are described.

This explains the limitations.
Of particular interest to me was the sampling for data collection.
The survey was conducted as an online survey. As such, it was assumed that the respondents had a certain level of IT literacy. This implies that the respondents have a certain level of information gathering ability. Therefore, the impact of people with low information gathering ability may not be taken into account.
This point has also been shown for access to the Internet. However, I think what is important is not the access to the Internet itself, but the bias due to the accessibility of information and the bias due to the reliability of the information source.

Thank you for your comment. We agree that individuals without social networks and who use email might not have come across the questionnaire, creating accessibility bias. We’ve updated the limitations to refer to those biases:

“Additionally, homeless and users with limited internet access are likely underrepresented and users without a social network profile or email account and/or with limited digital literacy or IT skills might not even have come across the questionnaire.“

Round 2

Reviewer 1 Report

Thank you for responding to my concerns. 

Please make it clear that "Questions about sociodemographic characteristics were asked variables were presented at the end of the questionnaire." I currently don't see this clarification in the manuscript.

Author Response

Thank you for noticing. We've added the sentence in the methodology section:

"Questions about sociodemographic characteristics were presented at the end of the questionnaire."